# Biochemical monitoring after initiation of aldosterone antagonist therapy in users of renin–angiotensin system blockers: a UK primary care cohort study

Sarah-Jo Sinnott,[1] Kathryn E Mansfield,[1] Morten Schmidt,[1,2,3] Krishnan Bhaskaran,[1] Liam Smeeth,[1] Dorothea Nitsch,[1] Laurie A Tomlinson[1]

[1]Department of Non-Communicable Disease Epidemiology, London School of Hygiene and Tropical Medicine, London, UK
[2]Department of Clinical Epidemiology, Aarhus University Hospital, Aarhus, Denmark
[3]Department of Cardiology, Regional Hospital West Jutland, Herning, Denmark

**Correspondence to**
Dr Sarah-Jo Sinnott;
sarah-jo.sinnott@lshtm.ac.uk

## ABSTRACT

**Objective** To determine the frequency of biochemical monitoring after initiation of aldosterone antagonists(AA) in patients also using angiotensin-converting enzyme inhibitors/angiotensin receptor blockers (ACEI/ARB).

**Setting** UK primary care.

**Participants** ACEI/ARB users who initiated AA between 2004 and 2014.

**Outcomes** We calculated the proportions with: (1) biochemical monitoring ≤2 weeks post initiation of AA, (2) adverse biochemical values ≤2 months (potassium ≥6 mmol/L, creatinine ≥220 µmol/L and ≥30% increase in creatinine from baseline) and (3) discontinuers of AA in those with an adverse biochemical value. We used logistic regression to study patient characteristics associated with monitoring and adverse biochemical values.

**Results** In 10 546 initiators of AA, 3291 (31.2%) had a record of biochemical monitoring ≤2 weeks post initiation. A total of 2.0% and 2.7% of those with follow-up monitoring within 2 months of initiation experienced potassium ≥6 mmol/L and creatinine ≥220 µmol/L, respectively, whereas 13.5% had a ≥30% increase in creatinine. Baseline potassium (OR 3.59, 95% CI 2.43 to 5.32 for 5.0–5.5 mmol/L compared with <5.0 mmol/L) and estimated glomerular filtration rate 45-59 ml/min/1.73 m$^2$ (OR 2.06, 95% CI 1.26 to 3.35 compared with ≥60 ml/min/1.73 m$^2$) were independently predictive of potassium ≥6 mmol/L. Women and people with diabetes had higher odds of ≥30% increase in creatinine.

**Conclusion** Less than one-third of patients taking ACEI/ARB had biochemical monitoring within 2 weeks of initiating AAs. Higher levels of monitoring may reduce adverse biochemical events.

## INTRODUCTION

Angiotensin-converting enzyme inhibitors or angiotensin receptor blockers (ACEI/ARB) and aldosterone antagonists (AA), such as spironolactone and eplerenone, are frequently used in combination. They provide reductions in morbidity and mortality for patients with heart failure[1] and

### Strengths and limitations of this study

► This is a population cohort study, based on electronic health records from UK primary care, examining whether users of renin–angiotensin system blockade who commence aldosterone antagonists (AAs) have appropriate biochemical monitoring after initiation of AA.
► The population was not restricted by indication for therapy.
► Those who were hospitalised prior to or immediately after initiating AA may have had missing data for test results in primary care data. In a sensitivity analysis, we used primary care data linked to hospital data to assess monitoring in a population that was not hospitalised. We found similar rates of monitoring and adverse events compared with the main analysis.

reductions in blood pressure for patients with resistant hypertension.[2] However, users of these drugs are at risk of acute kidney injury,[3] hyperkalaemia and hyperkalaemia-associated mortality.[4]

The occurrence of adverse events associated with combined ACEI/ARB and AA use was highlighted in early 2016 in the UK with the publication of a drug safety notice from the Medicines and Healthcare products Regulatory Agency (MHRA). It reported on the increasing incidence of life-threatening hyperkalaemic adverse events in patients prescribed ACEI/ARB and spironolactone.[5]

To help avoid adverse events after initiation of AA, biochemical parameters should be monitored.[6] At present, the National Institute of Health and Care Excellence (NICE) practical guidelines for heart failure recommend testing for potassium, creatinine and estimated glomerular filtration rate (eGFR)

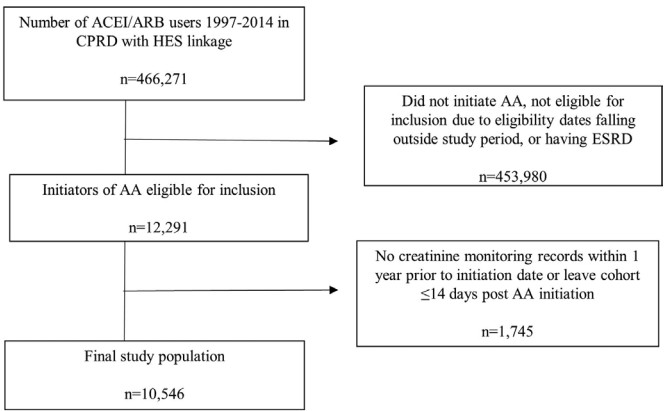

**Figure 1** Flow chart demonstrating cohort selection. AA, aldosterone antagonist; ACEI/ARB, angiotensin-converting enzyme inhibitor/angiotensin receptor blocker; CPRD, Clinical Practice Research Datalink; ESRD, end-stage renal disease; HES, Hospital Episode Statistics.

after 1 week; 1, 2, 3 and 6 months; and 6 monthly thereafter, following initiation of AA in heart failure.[7] These guidelines recommend stopping the AA if potassium is ≥6 mmol/L and if creatinine is ≥220 µmol/L. NICE guidelines for hypertension state that testing for sodium, potassium and renal function should occur within 1 month after initiation of AA and as required thereafter.[8]

It is not known how well these guidelines are adhered to in the UK. Previous evidence on blood testing during AA treatment in the UK is historic, restricted to one geographical region and did not specifically assess adherence to guideline-recommended blood testing.[9] Data from North America suggest that recommended blood testing occurs in less than 50% of patients.[3 10 11] Poor monitoring of patients taking these drug combinations, as well as increasing use among patients at high risk of adverse outcomes, may help to explain the increased occurrence of hyperkalaemic events as reported by the MHRA.

Therefore, among a large, recent cohort of users of ACEI/ARB who initiated AA, we sought to examine patterns of blood testing and the occurrence of hyperkalaemia and renal impairment. Our aims were to determine: (1) the proportion of patients initiating an AA who had testing within 2 weeks of initiation, (2) the patient characteristics associated with testing, (3) the proportion of patients who had adverse biochemical values post initiation of AA and the proportion that then discontinued the AA and (4) the patient characteristics associated with adverse biochemical values.

## METHODS

### Data

The Clinical Practice Research Datalink- (CPRD) is a nationally representative repository of deidentified electronic medical records from primary care in the UK. It holds data on demographics, health-related behaviours, test results, diagnoses, referrals and prescriptions for more than 11 million people with research-quality data.[12] It is one of the largest databases of longitudinal medical records from primary care globally and has been extensively validated.[12 13]

For this study, we used CPRD data linked to Hospital Episodes Statistics (HES). This linkage is possible for 60% of English practices contributing to the CPRD database. The HES database provides data on the primary diagnosis for a hospital admission, as well as other diagnoses and procedures carried out during that admission. The linkage thus provides a more complete picture of comorbidities, improves the accuracy of timing and, in this study, allowed the conduct of sensitivity analyses in those without hospital admissions.[14]

### Population

We identified a cohort of patients with HES-linked CPRD data aged ≥18 years, who initiated ACEI/ARB treatment between 1 April 1997 and 31 March 2014.[15] We identified continuous courses of ACEI/ARB therapy by allowing for a 90-day gap between the end date of one prescription and the start of the next consecutive prescription (to allow for stockpiling and medications prescribed in secondary care). Among this cohort, we identified people who subsequently became new users of AA in the period 2004–2012. We chose 2004 as the study start year to coincide with the roll-out of the Quality and Outcomes Framework (QOF), a system of remuneration to general practitioners (GPs) which encouraged the standardisation of primary medical services. The introduction of QOF resulted in more recording of renal function test results due to the incentivisation of creatinine testing among people with diabetes in 2004 and the establishment of a practice Chronic Kidney Disease (CKD) register with the introduction of eGFR reporting in 2006.[16]

New use of AA was defined as no use of AA in the year prior to first prescription. We then restricted the population to those with a record for creatinine monitoring in the year prior to AA initiation. We assumed that those with creatinine test results prior to and after AA initiation were also tested for potassium. If values for potassium were not present, they were treated as missing. This approach avoided exclusion of patients whose blood sample may have been haemolysed, resulting in potassium value not reported. Patients with Read codes for end-stage renal disease and eGFR values corresponding with CKD stage 5 prior to cohort entry were excluded. Patients were eligible for follow-up until the earliest of death, transfer out of practice, last data collection or end of the study (March 2014).

### Covariates

We obtained information for all patients on age, gender, calendar time of AA initiation (2004–2006, 2007–2009 and 2010–2014) and lifestyle factors. The closest records to AA initiation date were used for determining smoking, alcohol and body mass index status using existing algorithms.[17] We extracted data on cardiovascular

comorbidities and diabetes using data from both CPRD and HES. We calculated baseline eGFR using the most recent creatinine value from CPRD data prior to AA initiation and the CKD Epidemiology Collaboration equation.[18] CPRD prescribing data were used to extract data on baseline medication use. Baseline potassium values were categorised as: <5, 5–5.5 and >5.5 mmol/L.

## Outcomes and statistical analyses

Several guidelines recommend time periods for blood testing after initiation of AA (online supplementary table 1). We assessed whether monitoring occurs in line with the current NICE recommendations, with some modification. NICE practical guidelines for heart failure recommend blood testing within 1 week after AA initiation.[7] However, we calculated the proportion of AA initiators who had blood testing within 2 weeks of AA initiation to accommodate the practical challenges of immediate follow-up testing faced in clinical practice. In additional analyses, we also calculated the proportion of people who had testing (1) within 7 days post initiation of AA, (2) within 1, 2, 3, 6 and 12 months post initiation of AA and (3) on all recommended monitoring occasions. Guidelines offer a framework for clinicians on how to treat and manage patients, but adherence to guidelines is not obligatory and deviations may just reflect individualised care. Nonetheless, using clinical guidelines to frame this analysis provides a series of time points against which we can quantify frequency of monitoring for patients using combinations of AA and ACEI/ARB.

We used thresholds set out by the NICE practical guidelines for heart failure to calculate the proportion of patients who had an adverse biochemical value on their first blood test within 2 months of initiation.[7] Hyperkalaemia was a potassium of ≥6 mmol/L, and an adverse creatinine value indicating renal dysfunction was defined as creatinine ≥220 µmol/L. We chose 2 months to accord with monitoring periods in previous clinical trials[19] and also because at the outset, we expected that a majority of patients would have testing within this time frame. We then calculated the proportion of those with adverse biochemical values who discontinued the AA. We used a conservative definition of discontinuation to prevent misclassification of people who had blood tests at the beginning of a median length prescription (28 days). Therefore, we classified discontinuation as no further prescription of AA beyond 30 days after the first postinitiation blood test, that is, when the end date of the course of AA therapy occurred before the first blood test date plus 30 days.

Some guidelines for initiation of AA refer to an absolute level of creatinine to indicate renal dysfunction of concern and when stopping the AA should be considered (online supplementary table 1). However, absolute values of creatinine reflect substantially different levels of eGFR depending on the age, gender and ethnicity of the patient. A proportional change in creatinine or eGFR is recommended by the NICE guidelines for CKD to indicate significant change in renal function for patients who initiate ACEI/ARB.[20] This measure is therefore clinically familiar and more closely indicates changes in renal function that may be associated with a new drug. Therefore, we also calculated the number of people who experienced a ≥30% relative increase in creatinine from baseline as an adverse biochemical finding. We used the most recent values for creatinine within 1 year prior to AA initiation and creatinine values on the first blood test within 2 months post initiation of AA to calculate the relative change.

To assess patient level characteristics associated with testing within 2 weeks (vs not having testing within 2 weeks), we used logistic regression with robust standard errors to adjust for correlations between patients within practices.[10] The crude model adjusted for age and gender only, whereas the fully adjusted model adjusted for age, gender, eGFR category, cardiovascular comorbidities, diabetes, potassium and calendar time all meaured at baseline. We did not include ethnicity in fully adjusted models due to missing data for approximately 50% of the population.[21] We also used logistic regression to assess patient characteristics associated with an adverse biochemical value (vs not having an adverse biochemical value), with adjustments for the same variables as in the prior model.

In a sensitivity analysis, we assessed the proportion of patients receiving blood tests among patients who were not hospitalised in the 30-day period prior to or post initiation of AA. Data on laboratory tests are not available in the HES database, and test results may not always be sent from the hospital to the GP practice. Therefore, this analysis excluded patients who had blood tests as inpatients, which would contribute to an apparently low observed level of testing in primary care.

We used Stata V.14 for all analyses.[22]

The protocol for this study was approved by the London School of Hygiene and Tropical Medicine Ethics Committee (no 6536) and the Independent Scientific Advisory Committee for MHRA (no 16_025A).

## RESULTS

From 466 271 continuous users of ACEI/ARB between 1997 and March 2014, 12 291 people initiated an AA between 2004 and 2014 and were eligible for inclusion. After further exclusions, 10 546 were ultimately included in the cohort (figure 1). The population was 41% women and had a mean age of 71.8 years (SD 12.9) (table 1). Mean baseline serum potassium was 4.4 mmol/L (SD 0.7). Approximately one-fifth of the population had an eGFR <30 ml/min/1.73 m$^2$ (table 1). Spironolactone was the drug given for 9917/10 546 (94%) of those initiating an AA, with the remainder initiating eplerenone.

## Testing within 2 weeks of AA initiation

Within 2 weeks post initiation of AA, 31.2% of the cohort had blood testing (table 1). Approximately 64% of the

**Table 1** Characteristics of patients taking ACEI/ARB who initiate AA in UK primary care 2004–2014, by monitoring groups

| | Population | Monitoring ≤2 weeks | No monitoring ≤2 weeks |
|---|---|---|---|
| Total number | 10 546 (100) | 3291 (100) | 7255 (100) |
| Female sex | 4348 (41.2) | 1326 (40.3) | 3022 (41.6) |
| Age (years) | | | |
| <50 | 656 (6.2) | 138 (4.2) | 518 (7.1) |
| 50–59 | 1188 (11.3) | 314 (9.5) | 874 (12.1) |
| 60–64 | 957 (9.1) | 275 (8.4) | 682 (9.4) |
| 65–69 | 1227 (11.6) | 391 (11.9) | 836 (11.5) |
| 70–74 | 1513 (14.4) | 502 (15.3) | 1011 (13.9) |
| 75–79 | 1708 (16.2) | 552 (16.8) | 1156 (15.9) |
| ≥80 | 3297 (31.3) | 1119 (34.0) | 2178 (30.0) |
| Ethnicity | | | |
| White | 4687 (44.4) | 1476 (44.9) | 3211 (44.3) |
| South Asian | 119 (1.1) | 33 (1.0) | 86 (1.2) |
| Black | 101 (1.0) | 23 (0.7) | 78 (1.1) |
| Mixed/other | 58 (0.6) | 11 (0.3) | 47 (0.7) |
| Missing/not stated | 5581 (52.9) | 1748 (53.1) | 3833 (52.8) |
| Smoking | | | |
| Non-smoker | 2693 (25.5) | 835 (25.4) | 1858 (25.6) |
| Current smoker | 1505 (14.3) | 405 (12.3) | 1100 (15.2) |
| Ex-smoker | 6333 (60.1) | 2047 (62.2) | 4286 (59.1) |
| Missing | 15 (0.1) | * | 11 (0.2) |
| Body mass index (kg/m$^2$) | | | |
| Underweight (<18.5) | 191 (1.8) | 69 (2.1) | 122 (1.7) |
| Healthy weight (18.5–24.9) | 2613 (24.8) | 823 (25.0) | 1790 (24.7) |
| Overweight (25–29.9) | 3335 (31.6) | 1020 (31.0) | 2315 (31.9) |
| Obese (≥30) | 4407 (41.8) | 1379 (41.9) | 3028 (41.7) |
| Missing | * | * | * |
| Alcohol | | | |
| Non-drinker | 1268 (12.0) | 360 (10.9) | 908 (12.5) |
| Current drinker | 7194 (68.2) | 2308 (70.1) | 4886 (67.4) |
| Ex-drinker | 1451 (13.8) | 447 (13.6) | 1004 (13.8) |
| Missing | 633 (6.0) | 176 (5.4) | 457 (6.3) |
| Comorbidities | | | |
| Hypertension | 7928 (75.2) | 2496 (75.8) | 5432 (74.9) |
| IHD | 6042 (57.3) | 1919 (58.3) | 4123 (56.8) |
| Heart failure | 6171 (58.5) | 1983 (60.3) | 4188 (57.7) |
| Arrhythmia | 4495 (42.6) | 1493 (45.4) | 3002 (41.4) |
| Diabetes | 3252 (30.8) | 1043 (31.7) | 2209 (30.5) |
| Peripheral arterial disease | 946 (9.0) | 326 (9.9) | 6202 (8.6) |
| Medication use | | | |
| Calcium channel blocker | 5043 (47.8) | 1578 (48.0) | 3465 (47.8) |
| Beta blocker | 6123 (58.1) | 1976 (60.0) | 4147 (57.2) |
| Thiazide diuretic | 4354 (41.3) | 1384 (42.1) | 2970 (40.9) |
| Loop diuretic | 7447 (70.6) | 2490 (75.7) | 4957 (68.3) |
| eGFR category (mL/min/1.73 m$^2$)† | | | |
| ≥60 | 5803 (55.0) | 1665 (50.6) | 4138 (57.0) |

**Table 1** Continued

| | Population | Monitoring ≤2 weeks | No monitoring ≤2 weeks |
|---|---|---|---|
| 45–59 | 2668 (25.3) | 864 (26.3) | 1805 (24.9) |
| 30–44 | 1666 (15.8) | 591 (18.0) | 1075 (14.8) |
| 15–29 | 409 (3.9) | 171 (5.2) | 237 (3.3) |
| Calendar time | | | |
| 2004–2006 | 2374 (22.5) | 645 (19.6) | 1729 (23.8) |
| 2007–2009 | 3029 (28.7) | 943 (28.7) | 2086 (28.8) |
| 2010–2014 | 5143 (48.8) | 173 (51.8) | 3440 (47.4) |
| Biochemical parameters at baseline‡ | | | |
| Creatinine (μmol/L) (mean (SD)) | 99.1 (30.7) | 102.7 (33.4) | 97.4 (29.3) |
| Potassium (mmol/L) (mean (SD)) | 4.4 (0.7) | 4.4 (0.9) | 4.4 (0.6) |
| Potassium (mmol/L) | | | |
| <5.0 | 8955 (84.9) | 2779 (84.4) | 6176 (85.1) |
| 5.0–5.5 | 1136 (10.8) | 367 (11.2) | 769 (10.6) |
| >5.5 | 252 (2.4) | 78 (2.4) | 174 (2.4) |
| Missing | 203 (1.9) | 67 (2.0) | 136 (1.9) |

Data shown are column number (%), except where mean (SD) is indicated.
*Refers to negligible.
†Calculated from most recent creatinine measurement within 12 months before AA initiation.
‡Baseline biochemical parameters were the values closest to AA initiation in the preceding year.
AA, aldosterone antagonist; ACEI/ARB, angiotensin-converting enzyme inhibitor/angiotensin receptor blocker; eGFR, estimated glomerular filtration rate; IHD, ischaemic heart disease.

population had testing within 2 months of initiation, whereas 95% had blood tests within 1 year (table 2). We conducted a sensitivity analysis among the 5787 individuals not admitted to hospital within 30 days prior or post initiation of AA (54.9% of the whole cohort). Among these patients, a similar proportion (32.7%) had follow-up monitoring within 2 weeks (online supplementary table 2). In the main population, the proportion of people having six tests within a year (roughly equating to optimal guideline-recommended testing within 7 days and 1, 2, 3, 6 and 12 months) testing was approximately 1%.

### Factors associated with testing within 2 weeks of AA initiation

Women and younger patients had lower odds for blood testing within 2 weeks of AA initiation (table 3). Those initiating AA after 2007 had higher odds of blood testing compared with patients initiating in 2004–2006. Those with high baseline potassium values did not have higher odds for receiving blood testing. No substantial differences were seen when the analysis was restricted to non-hospitalised patients although the increased odds of testing among patients with reduced baseline renal function was reduced, suggesting that there were disproportionate admissions among this group (online supplementary table 3).

### Adverse biochemical values

Of those with follow-up monitoring within 2 months of AA initiation (n=6520), 2.0% had hyperkalaemia, 2.7% a creatinine ≥220 μmol/L and 13.5% a ≥30% increase in creatinine from baseline, on their first blood test (table 4). Of those with creatinine ≥220 μmol/L, 29 people (16%) had creatinine ≥220 μmol/L at baseline. Approximately half of patients with hyperkalaemia discontinued AA within 30 days of the blood test (table 4). Less than one-third of those with a ≥30% increase in creatinine discontinued AA, compared with 43% of those with a postinitiation creatinine ≥220 μmol/L. Among non-hospitalised patients, the proportions with adverse biochemical values were lower, whereas the proportion of people discontinuing the AA with each

**Table 2** Proportion of patients taking ACEI/ARB who initiate AA in UK primary care 2004–2014 with blood tests at several time points post initiation of AA

| 7 days | 14 days | 1 month | 2 months | 3 months | 6 months | 1 year |
|---|---|---|---|---|---|---|
| 1674/10 613 | 3291/10 546 | 5210/10 431 | 6520/10 258 | 7229/10 145 | 8389/9962 | 9294/9799 |
| 15.8% | 31.2% | 49.9% | 63.5% | 71.3% | 84.2% | 94.9% |

The denominator changes due to patients exiting the cohort.
AA, aldosterone antagonist; ACEI/ARB, angiotensin-converting enzyme inhibitor/angiotensin receptor blocker.

adverse biochemical value was similar (online supplementary table 4).

## Factors associated with adverse biochemical values

In fully adjusted logistic regression models, baseline eGFR categories <60 ml/min/1.73 m$^2$ and potassium ≥5 mmol/L were independently associated with hyperkalaemia (table 5). Women had lower odds than men for a postinitiation creatinine ≥220 µmol/L but had a higher odds of a ≥30% increase in creatinine. In sensitivity analyses of the non-hospitalised cohort, similar associations between age and sex and the occurrence of adverse biochemical values were observed (online supplementary table 5).

## DISCUSSION

In this cohort study of 10 546 users of ACEI/ARB, less than a third had follow-up blood testing within 2 weeks of AA initiation. Less than two-thirds had blood testing within 2 months, and of these, 2% developed severe hyperkalaemia, and a similar proportion developed an absolute value of creatinine at which some guidelines recommend that AA cessation is considered. In the same time frame, 13.5% of the cohort developed a ≥30% increase in serum creatinine from baseline.

## Strengths and weaknesses

We used a linked primary and secondary care dataset to ensure complete information on comorbid diagnoses and to allow sensitivity analyses for those without hospital admissions. These data had high levels of completeness for baseline renal function and comorbidities allowing the majority of ACEI/ARB users initiating AA to be included in the cohort. A limitation of our study was that we could only examine adverse biochemical values for those who had a record of subsequent blood testing in primary care. The risk of adverse values may have been different in patients who were not tested or were tested in secondary care. Lack of access to blood test results from secondary care could have created a bias towards missing data among the sickest patients and an underestimation of the rate of adverse biochemical values. However, restricting the analysis to individuals without any record of a hospitalisation in the 30-day period prior to and after AA initiation did not substantially change our results. A further limitation is that for quantifying baseline renal function, we could not include an indicator for Afro-Caribbean ethnicity in the eGFR calculation. However, we expect the

**Table 3** Association between patient characteristics and biochemical monitoring within 2 weeks post-initiation of AA

| | OR (95% CI) | |
| --- | --- | --- |
| | Age and sex adjusted | Fully adjusted |
| Male | Ref | Ref |
| Female | 0.90 (0.83 to 0.98) | 0.90 (0.82 to 0.98) |
| **Age (years)** | | |
| <50 | 0.54 (0.43 to 0.67) | 0.58 (0.46 to 0.73) |
| 50–59 | 0.72 (0.61 to 0.84) | 0.78 (0.66 to 0.91) |
| 60–64 | 0.81 (0.68 to 0.96) | 0.86 (0.72 to 1.03) |
| 65–69 | 0.94 (0.81 to 1.09) | 0.96 (0.83 to 1.12) |
| 70–75 | Ref | Ref |
| 76–79 | 0.90 (0.83 to 0.98) | 0.95 (0.82 to 1.10) |
| ≥80 | 1.05 (0.92 to 1.20) | 0.97 (0.85 to 1.11) |
| **eGFR category (mL/min/1.73 m$^2$)** | | |
| ≥60 | Ref | Ref |
| 45–59 | 1.10 (0.99 to 1.23) | 1.13 (1.01 to 1.27) |
| 30–44 | 1.25 (1.10 to 1.42) | 1.29 (1.14 to 1.46) |
| 15–29 | 1.65 (1.33 to 2.06) | 1.74 (1.40 to 2.16) |
| **Comorbidities** | | |
| Hypertension | 1.03 (0.93 to 1.13) | 0.98 (0.88 to 1.09) |
| IHD | 1.00 (0.92 to 1.09) | 0.98 (0.89 to 1.07) |
| Heart failure | 1.05 (0.96 to 1.15) | 1.01 (0.92 to 1.11) |
| Arrhythmia | 1.09 (1.00 to 1.19) | 1.07 (0.98 to 1.18) |
| Diabetes | 1.06 (0.97 to 1.16) | 1.05 (0.95 to 1.15) |
| Peripheral arterial disease | 1.12 (0.98 to 1.27) | 1.08 (0.93 to 1.24) |
| **Calendar time** | | |
| 2004–2006 | Ref | Ref |
| 2007–2009 | 1.22 (1.07 to 1.38) | 1.21 (1.07 to 1.38) |
| 2010–2014 | 1.34 (1.17 to 1.53) | 1.35 (1.18 to 1.55) |
| **Baseline potassium (mmol/L)** | | |
| <5.0 | Ref | Ref |
| 5.0–5.5 | 1.05 (0.91 to 1.20) | 0.98 (0.86 to 1.12) |
| >5.5 | 0.98 (0.73 to 1.30) | 0.90 (0.68 to 1.20) |

Fully adjusted: adjusted for age, gender, eGFR category, hypertension, heart failure, ischaemic heart disease, arrhythmias, diabetes, peripheral arterial disease, baseline potassium and calendar time. Further adjustment for lifestyle covariates made marginal difference to all results, thus these variables are not included in models shown.
AA, aldosterone antagonist; eGFR, estimated glomerular filtration rate; IHD, ischaemic heart disease.

**Table 4** Proportion with adverse biochemical values on testing within 2 months post initiation of AA and number subsequently discontinuing AA

| | Hyperkalaemia (≥6 mmol/L)* | Creatinine ≥220 µmol/L | ≥30% increase in creatinine |
| --- | --- | --- | --- |
| Number with adverse biochemical values† | 128/6373 (2.0%) | 177/6520 (2.7%) | 877/6520 (13.5%) |
| Number discontinuing AA‡ | 68/128 (53.1%) | 76/177 (42.9%) | 251/877 (28.6%) |

*Missing data for 147 people (2.3%) for first follow-up potassium value.
†Serum potassium and creatinine values on first blood test within 2 months of AA initiation.
‡Discontinuation defined as no further prescriptions of AA after blood test plus 30 days.
AA, aldosterone antagonist.

**Table 5** Associations between patient characteristics and adverse biochemical values after AA initiation

| | OR (95% CI) | | |
|---|---|---|---|
| | Hyperkalaemia*† (≥6 mmol/L) | Creatinine*≥220 µmol/L | ≥30% increase in creatinine* |
| Female | 1.04 (0.70 to 1.54) | 0.34 (0.23 to 0.51) | 1.47 (1.25 to 1.73) |
| Age (years) | | | |
| <50 | 0.15 (0.02 to 1.08) | 0.96 (0.32 to 2.84) | 0.34 (0.21 to 0.55) |
| 50–59 | 0.29 (0.10 to 0.86) | 0.80 (0.32 to 2.84) | 0.45 (0.32 to 0.65) |
| 60–64 | 0.60 (0.28 to 1.31) | 1.17 (0.53 to 2.56) | 0.69 (0.49 to 0.96) |
| 65–69 | 0.83 (0.43 to 1.59) | 0.36 (0.16 to 0.83) | 0.64 (0.49 to 0.85) |
| 70–75 | Ref | Ref | Ref |
| 76–79 | 0.75 (0.43 to 1.33) | 0.71 (0.42 to 1.20) | 0.94 (0.75 to 1.18) |
| 80+ | 0.69 (0.40 to 1.19) | 0.50 (0.30 to 0.83) | 1.10 (0.87 to 1.39) |
| eGFR category (mL/min/1.73 m$^2$) | | | |
| ≥60 | Ref | Ref | Ref |
| 45–59 | 2.06 (1.26 to 3.35) | 2.80 (1.22 to 6.39) | 0.81 (0.66 to 0.99) |
| 30–44 | 2.32 (1.31 to 4.11) | 20.83 (9.73 to 44.58) | 0.82 (0.67 to 1.00) |
| 15–29 | 3.62 (1.60 to 8.22) | 248.0 (117.2 to 524.7) | 1.01 (0.72 to 1.40) |
| Comorbidities | | | |
| Hypertension | 1.27 (0.76 to 2.14) | 0.93 (0.61 to 1.41) | 1.18 (0.98 to 1.41) |
| IHD | 0.67 (0.45 to 1.00) | 0.60 (0.43 to 0.85) | 0.95 (0.82 to 1.11) |
| Heart failure | 0.77 (0.53 to 1.14) | 0.97 (0.67 to 1.41) | 1.12 (0.96 to 1.31) |
| Arrhythmia | 1.00 (0.68 to 1.46) | 0.81 (0.57 to 1.15) | 1.11 (0.96 to 1.28) |
| Diabetes | 1.21 (0.83 to 1.76) | 1.11 (0.80 to 1.55) | 1.18 (1.01 to 1.38) |
| Peripheral arterial disease | 0.82 (0.44 to 1.51) | 1.04 (0.64 to 1.68) | 1.09 (0.85 to 1.40) |
| Calendar time | | | |
| 2004–2006 | Ref | Ref | Ref |
| 2007–2009 | 1.14 (0.65 to 2.01) | 1.36 (0.84 to 2.22) | 0.97 (0.80 to 1.18) |
| 2010–2014 | 1.08 (0.64 to 1.83) | 0.93 (0.58 to 1.51) | 0.95 (0.78 to 1.15) |
| Baseline potassium (mmol/L) | | | |
| <5.0 | Ref | n/a | n/a |
| 5.0–5.5 | 3.59 (2.43 to 5.32) | | |
| >5.5 | 3.29 (1.57 to 6.88) | | |

Only fully adjusted model is shown for clarity. Adjusted for age, gender, eGFR category, hypertension, heart failure, ischaemic heart disease, arrhythmias, diabetes, peripheral arterial disease, baseline potassium (hyperkalaemia model only) and calendar time. Further adjustment for lifestyle covariates made marginal difference to all results and is not included.

n=6520 except †missing data for 147 people for first follow-up potassium value.

*Adverse serum potassium and creatinine values on first blood test within 2 months of AA initiation.

AA, aldosterone antagonist; eGFR, estimated glomerular filtration rate; IHD, ischaemic heart disease.

impact of this to be small given that only 3% of the English population is of Afro-Caribbean ethnicity, and the proportion is lower in the older age group of this study.[23]

To be pragmatic, we quantified testing for all patients against the NICE heart failure practical guidelines, although not all patients will have had heart failure as the indication for treatment. We chose not to limit to patients coded to have heart failure because comorbidity coding may not be complete, because coding may be related to routine care received[24] and also because there is a growing evidence base for use of AA in resistant hypertension.[2] In addition, even for those with coded heart failure, we did not have details of patient's echocardiographic findings so could not restrict to those with impaired left ventricular function, although this is the group in whom the benefits of treatment despite the risk of worsening renal function are clearest.[25]

A strength of our study is that as well as examining the biochemical values of concern noted in the current NICE practical guidelines for heart failure, we also assessed the incidence of a ≥30% increase in creatinine after initiation of AA. This outcome is clinically familiar from monitoring after ACEI/ARB initiation but is not discussed in the practical guidelines as a parameter in the benefit/risk evaluation of continuing AA therapy. The clinical significance of such deteriorations in this

context is not clear and may indicate favourable haemo-dynamic changes associated with improved outcomes.[6] However, worsening of renal function has been shown to have prognostic significance after initiation of ACEI/ARB treatment,[26] particularly in the context of heart failure with preserved ejection fraction.[25]

## Comparison with other studies

In a study of patients with heart failure receiving health-care through the Veterans Affairs health insurance programme in the USA between 2003 and 2013, blood testing was carried out in 42% of patients within 2 weeks of initiation of AA.[10] Two other studies of patients with heart failure carried out in North American settings found slightly higher monitoring rates for AA initiators; 53.5% within 7 days of initiation[3] and 44.6% within 10 days of initiation.[11] The contrast between our results and the results of North American research is noteworthy because it has been suggested in the past that monitoring in the UK is more frequent than in Canada.[4 9] However, prior research on the frequency of monitoring in the UK was carried out in one geographical area and did not directly assess concordance with guideline-recommended testing.[9]

Previous research examining patient characteristics associated with blood testing within 13 months of AA initiation found similar results to our study; that males, older patients and those with CKD had higher odds of receiving blood tests.[27] While diabetes was weakly associated with testing in our study (OR 1.2 in sensitivity analysis of non-hospitalised patients), the association was less pronounced than in prior research (OR 1.6).[27]

We found that 2% of those initiating an AA experienced hyperkalaemia (potassium ≥6 mmol/L) on the first blood testing event within 2 months of AA initiation. This compares to a rate of 2% found in Randomized Aldactone Evaluation Study and 5.5% in the Eplerenone Post–Acute Myocardial Infarction Heart Failure Efficacy and Survival Study, both clinical trials of AA in patients with heart failure.[1 28] In observational studies, rates of hyperkalaemia ranging from 6%[29] to 10%[30] have been reported. Such populations are said to be more reflective of the typical user of AA in clinical practice than in clinical trials, and thus, the rates are thought to be more realistic.[4] The disparity between our results for hyperkalaemia and those found in other observational research may have stemmed from unknown rates of adherence in this unselected population, from differences in doses used[6] or from differences in methodology. For example, we analysed potassium values only on the first blood test within 2 months, whereas prior studies examined peak potassium values within 3 months of initiation[29] and peak values within the individuals' entire treatment period.[30] As time since initiation increases, it is increasingly likely that blood tests in primary care are performed due to an intercurrent illness or other event that may itself have contributed to hyperkalaemia or reduced renal function. Thus,

we opted to look at the first blood testing event within 2 months to examine as much as possible a causal association between the initiation of AA and the adverse biochemical value. In line with previous literature, we found that baseline potassium value was a significant predictor of hyperkalaemia after initiation of AA[28 29 31] as was reduced renal function.[27–29]

As currently referenced in some guidelines (online supplementary table 1), using a threshold creatinine value as a safety indicator for renal function may not be an optimal measure given that absolute creatinine levels create an age, gender and ethnicity bias. Importantly we found that, consistent with a lower muscle mass, women had a lower odds of developing a creatinine ≥220 μmol/L initiation of AA in comparison to men, but a 1.5-fold higher odds of a ≥30% increase in creatinine. The clinical implications of higher odds of ≥30% increase in creatinine are unknown, but it is important to establish that there is not a discrepancy in related clinical outcomes. However, gender-related differences in adverse biochemical values were not a prespecified hypothesis so this finding must be considered hypothesis-generating.

## CONCLUSIONS AND CLINICAL IMPLICATIONS

We found that among patients prescribed an ACEI/ARB, less than one-third of those initiating an AA received follow-up blood tests within 2 weeks and less than two-thirds within 2 months. Approximately 2% of patients developed potassium ≥6 mmol/L or creatinine ≥220 μmol/L on the first blood test within 2 months, but 13.5% experienced a ≥30% increase in creatinine from baseline. Women had almost 50% higher odds of a ≥30% increase in creatinine than men. Baseline potassium levels >5.0 mmol/L and baseline eGFR categories <60 ml/min/1.73 m$^2$ were independently associated with a potassium value ≥6 mmol/L after AA initiation. Our results highlight the need for better adherence to monitoring guidelines, particularly for those in these high-risk groups and the importance of understanding the prognostic implications of changes in renal function after AA initiation.

**Contributors** LAT had the idea for the study and acquired data permissions. S-JS, KEM, MS, DN and LAT designed the study. S-JS, KEM and MS managed the data and established the cohort. S-JS did the analyses. All authors participated in the discussion and interpretation of the results. S-JS organised the writing and wrote the initial drafts. All authors critically revised the manuscript for intellectual content and approved the final version. S-JS is the guarantor.

**Funding** S-JS is supported by the Wellcome Trust Sir Henry Wellcome Fellowship (107340/Z/15/Z). LS is supported by the Wellcome Trust Senior Research Fellowship in Clinical Science (098504/Z/12/Z). LAT is supported by the Wellcome Trust Intermediate Clinical Fellowship (101143/Z/13/Z) which also funded KEM. KB is supported by the Sir Henry Dale Fellowship jointly funded by the Wellcome Trust and the Royal Society (107731/Z/15/Z). MS is supported by the AP Møller Foundation for the Advancement of Medical Science, Snedkermester Sophus Jacobsen og Hustru Astrid Jacobsens Fond and Christian og Ottilia Brorsons Rejselegat for yngre videnskabsmænd og-kvinder.

**Competing interests** S-JS, KEM, MS, DN and LAT have nothing to declare. LS reports grants from the Wellcome Trust and British Heart Foundation during the

conduct of the study; grants from the Wellcome Trust, Medical Research Council, National Institute for Health Research and European Union outside the submitted work; personal fees from GSK for advisory work unrelated to the submitted work; grant funding from GSK for academic research unrelated to the submitted work; acts as an unpaid steering committee chair for AstraZeneca for a randomised trial unrelated to the submitted work and is a trustee of the British Heart Foundation. KB declares grants from the Wellcome Trust, Royal Society, MRC, NIHR and BHF outside the submitted work.

**Ethics approval** The protocol for this study was approved by the London School of Hygiene and Tropical Medicine Ethics Committee (no 6536) and the Independent Scientific Advisory Committee (ISAC) for Medicines and Healthcare Products Regulatory Agency (no 16_025A).

**Provenance and peer review** Not commissioned; externally peer reviewed.

**Data sharing statement** No additional data available.

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
