## [Reviewer comments · BMJ Open]

ARTICLE DETAILS

TITLE (PROVISIONAL)	Laboratory monitoring after initiation of aldosterone antagonist therapy in users of renin-angiotensin system blockers: A UK primary care cohort study
AUTHORS	Sinnott, Sarah-Jo; Mansfield, Kathryn; Schmidt, Morten; Bhaskaran, Krishnan; Smeeth, Liam; Nitsch, Dorothea; Tomlinson, Laurie

VERSION 1 – REVIEW

REVIEWER	Bertram Pitt University of Michigan school of medicine, USA potential conflict of interests -Consultant -Bayer (finerenone), KDP* pharmaceuticals (a new non steroidal MRA) , Relypsa* (the potassium lowering drug Patiromer) *=stock options Patent pending -site specific delivery of eplerenone to the myocardium
REVIEW RETURNED	05-Jul-2017

GENERAL COMMENTS	This analysis of potassium monitoring in patients receiving a MRA adds to the growing literature in other areas of the world suggesting that physicians are not adequately monitoring serum K⁺ and renal function after instituting a MRA. While the results and conclusions based on this study are similar to those in the USA recently reported by cooper etal the information in the present analysis from the UK is important both to guide practice in the UK as well as to emphasize the need for better monitoring of serum K⁺ and renal function in those initialing a MRA . However , while I agree with further emphasis on monitoring of serum K⁺ and renal function post initiation of a MRA as well as discontinuing a MRA should hyperkalemia develop I would be less certain about the need to discontinue a MRA should creatinine increase since several studies have shown that an increase in serum creatinine is due to hemodynamic changes in renal perfusion and is reversible after discontinuing the MRA . You might wish to considering the article by Pitt , B and Rossignol , P - The safety of mineralocorticoid receptor antagonists (MRAs) in patients with heart failure . Expert opinion on drug safety 2016
--

REVIEWER	Franz Messerli University of Bern, Switzerland, Mount Sinai Icahn School of Medicine, New York and Jagiellonian University Krakow, Poland No Competing Interest
REVIEW RETURNED	12-Jul-2017

GENERAL COMMENTS	This is a well done, thorough paper providing the reader with a myriad of results. The main message, i.e. that less than one-third of patients taking ACEI/ARB had testing within 2 weeks of initiating AA, certainly is of concern. However, what we are missing are outcome data of such negligence. The paper merely documents changes in serum cosmetics such as potassium and creatinine but does not tell us whether or not these changes had any clinical consequences. In how many pts had AAs to be withdrawn because of AEs in monitored and unmonitored pts? In how many could it be started again at a lower dose? How many unexpected hospitalization occurred in the great majority of pts who were not monitored? What was the reason for such hospitalizations? Obviously the most important unanswered question is: Was there any outcome difference between pts who were and those who were not monitored? Since 94.9% were monitored within 12 months, does timing of monitoring make any difference? Unless some of these question are addressed one can make the devils advocate's point that monitoring just serves to relieve anxiety of the treating physician or to satisfy guideline requirements, but of course substantially increases cost, visits etc.
--

REVIEWER	Anastasios Lymperopoulos Nova Southeastern University, USA
REVIEW RETURNED	19-Jul-2017

GENERAL COMMENTS	1) Do you have any data on medication (in particular AA) adherence of your patient population? Please include this parameter (i.e. medication adherence) in your discussion of the potential reasons for the observed very low incidence of hyperkalemia among your study's patients. 2) Can you speculate on potential reasons why women had lower odds of getting tested within two weeks despite a higher risk of creatinine increase? Could differences in access to healthcare or other societal/behavioral factors be responsible for this gender-specific disparity? 3) The authors should (briefly) comment on the possible cost-effectiveness (or lack thereof) of such very early (<2 weeks post-AA initiation) biochemical monitoring of patients.
---

REVIEWER	Jordan Fulcher NHMRC Clinical Trials Centre, University of Sydney, Australia
REVIEW RETURNED	02-Aug-2017

GENERAL COMMENTS	This study is very well written. My only comments concern the way some of the findings have been interpreted. The finding that females had a lower odds of being tested in the first 2 weeks is valid, but an odds ratio of 0.9 only realistically highlights a small difference, and while the model presented was "fully adjusted" it would be preferable to also consider what differences in baseline characteristics there were between the genders to regard this finding as more definitive. I wonder if the fact that females had a higher odds of a >30% increase in creatinine, yet men had a higher odds of a serum creatinine >220umol/L relates to more than just body muscle mass and perhaps relates to gender differences in the severity of renal function at baseline. Equally, comparing cessation of treatment rates by gender would help to clarify the insinuation that females may be treated less expeditiously than men without clear justification. Consideration of these aspects would help to backup one of the main conclusions of the paper better. - I think it's also worth acknowledging that only 60% of the population had heart failure, and yet the guidelines being presented and used to guide this study's outcomes are focused predominantly on heart failure guidelines. If the conclusion is that there is a need for "better adherence to monitoring guidelines" then either only systolic heart failure patients should be in the population or there should be a discussion point that various guidelines recommend monitoring renal function in different ways. In the supplementary table most of the listed guidelines provide no recommendations for frequency of biochemical monitoring. - This leads onto the third point that I think a comment needs to be included regarding the fact that this data is descriptive, and that guideline recommendations are recommendations for clinicians, not mandated. If you are able to demonstrate higher rates of rehospitalisation or higher adverse clinical outcome rates associated with less frequent biochemical monitoring, then the conclusion would be more valid. Similarly a rise in creatinine to >220umol/L or >30% may not have any serious implications depending on baseline levels and clinical response. While I completely agree it's extremely useful to know how often people are being biochemically monitored and what their outcome rates are, none of these findings directly demonstrate negligent care, and may possibly (in a more optimistic way) reflect an individualised approach to patient care. - Finally its of little surprise that people with higher baseline potassium levels have higher rates of a potassium over 6, or that people with worse eGFRs have higher rates of creatinine>220 - the cutoffs are in guidelines, but are arbitrary. It would be also of research interest to consider the proportional declines in eGFR related to AA therapy, or to follow biochemical outcomes beyond the first post AA blood test to truly consider to clinical impacts of AA treatment for patients. I realise this may extend beyond the primary aims of the paper, but could be considered.
---

VERSION 1 – AUTHOR RESPONSE

Reviewer 1 Comments

Comment: This analysis of potassium monitoring in patients receiving a MRA adds to the growing literature in other areas of the world suggesting that physicians are not adequately monitoring serum K⁺ and renal function after instituting a MRA. While the results and conclusions based on this study are similar to those in the USA recently reported by Cooper et al the information in the present analysis from the UK is important both to guide practice in the UK as well as to emphasize the need for better monitoring of serum K⁺ and renal function in those initiating a MRA.

However, while I agree with further emphasis on monitoring of serum K⁺ and renal function post initiation of a MRA as well as discontinuing a MRA should hyperkalemia develop I would be less certain about the need to discontinue a MRA should creatinine increase since several studies have shown that an increase in serum creatinine is due to hemodynamic changes in renal perfusion and is reversible after discontinuing the MRA .

You might wish to consider the article by Pitt, B and Rossignol, P - The safety of mineralocorticoid receptor antagonists (MRAs) in patients with heart failure. Expert opinion on drug safety 2016

Response: Thank you for this useful reference. We have now added the citation to the introduction and the discussion (Line 12 page 5, Line 257 page 17, Line 242 Page 16). We agree that the prognostic importance of changes in creatinine in this context are uncertain and should not necessarily lead to drug cessation. We have altered the discussion to make this clear:

As well as examining the biochemical values of concern noted in current NICE practical guidelines for heart failure, we also assessed the incidence of a $\geq 30\%$ increase in creatinine after initiation of AA. This outcome is not discussed in the practical guidelines as a parameter for consideration of the benefits of continuing AA therapy, and indeed may indicate favourable haemodynamic changes associated with improved outcomes [6]. However, worsening of renal function has been shown to have prognostic significance after initiation of ACEI/ARB treatment, [23], particularly in the context of heart failure with preserved ejection fraction.[22]

Line 229 Page 15

We have added the following line to the discussion:

Even amongst those coded to have cardiac failure, we did not have details of patients echocardiographic findings so could not restrict to those with impaired left ventricular function, although this is the group in whom the benefits of treatment despite the risk of worsening of renal function are clearest.[22]

Reviewer 2 Comments

Comment: This is a well done, thorough paper providing the reader with a myriad of results. The main message, i.e. that less than one-third of patients taking ACEI/ARB had testing within 2 weeks of initiating AA, certainly is of concern.

Response: Thank you

Comment: However, what we are missing are outcome data of such negligence. The paper merely documents changes in serum cosmetics such as potassium and creatinine but does not tell us whether or not these changes had any clinical consequences.

- In how many pts had AAs to be withdrawn because of AEs in monitored and unmonitored pts?

- In how many could it be started again at a lower dose?
- How many unexpected hospitalization occurred in the great majority of pts who were not monitored?
- What was the reason for such hospitalizations?
- Obviously the most important unanswered question is: Was there any outcome difference between pts who were and those who were not monitored?
- Since 94.9% were monitored within 12 months, does timing of monitoring make any difference?

Unless some of these question are addressed one can make the devils advocate's point that monitoring just serves to relieve anxiety of the treating physician or to satisfy guideline requirements, but of course substantially increases cost, visits etc.

Response: We agree completely with this and are currently looking at the clinical consequences of these changes. To avoid making the initial paper too complex we aim to publish this work in two related papers as we have done with our previous work (Schmidt M et al, BMJ. 2017 Mar 9;356:j791 and Schmidt M et al, BMJ Open. 2017 Jan 9;7(1):e012818)

Reviewer 3 Comments

Comment: Do you have any data on medication (in particular AA) adherence of your patient population? Please include this parameter (i.e. medication adherence) in your discussion of the potential reasons for the observed very low incidence of hyperkalemia among your study's patients.

Response: We did not measure adherence to therapy in this study. This is methodologically challenging using prescribing data and is the subject of some of our related work. Rather, we conducted what can be regarded as an intention-to-treat analysis whereby the base population included those who initiated an AA. In the limitations section, we have listed some reasons that may explain the observed low incidence of hyperkalaemia amongst our study population: The disparity between our results for hyperkalaemia and those found in other observational research may have stemmed from unknown rates of treatment adherence in this unselected population, from differences in doses used[6], or from differences in methodology.

Line 269 Page 16

Comment: Can you speculate on potential reasons why women had lower odds of getting tested within two weeks despite a higher risk of creatinine increase? Could differences in access to healthcare or other societal/behavioral factors be responsible for this gender-specific disparity?

Response: As Reviewer 4 points out, the confidence intervals around this estimate are very close to 1 and, on reflection, we agree with that only limited emphasis can be placed on this finding. While it is possible that there are true gender related differences in access to healthcare or testing, we intend to consider this in more detail in our future related work.

We have therefore amended the Abstract to read:

Results

In 10,546 initiators of AA, 3,291 (31.2%) had a record of testing ≤ 2 weeks post initiation. A total of 2.0% and 2.7% of those with follow-up testing within 2 months of initiation experienced potassium ≥ 6 mmol/L and creatinine ≥ 220 μ mol/L respectively, while 13.5% had a $\geq 30\%$ increase in creatinine. Baseline potassium (OR 3.59, 95% CI 2.43-5.32 for 5.0-5.5mmol/L compared to < 5.0 mmol/L) and eGFR 45-59mls/min/1.73m² (OR 2.06, 95% CI 1.26–3.35 compared to ≥ 60 mls/min/1.73m²) were independently predictive of potassium ≥ 6 mmol/L. Women and people with diabetes had higher odds of $\geq 30\%$ increase in creatinine.

Conclusion

Less than one-third of patients taking ACEI/ARB had testing within 2 weeks of initiating aldosterone antagonists.

And the Discussion to read:

Using a threshold creatinine value as a safety indicator for renal function may not be an optimal measure given that absolute creatinine levels create an age, gender and ethnicity bias. Importantly we found that, consistent with a lower muscle mass, women had a lower odds of developing a creatinine $\geq 220\mu\text{mol/L}$ post AA initiation in comparison to men, but a 1.5 fold higher odds of a $\geq 30\%$ increase in creatinine. The clinical implications of higher odds of $\geq 30\%$ increase in creatinine are unknown but it is important to establish that there is not a discrepancy in related clinical outcomes. However, gender related differences in adverse biochemical values was not a pre-specified hypothesis so this finding must be considered hypothesis-generating.

Line 283 Page 17

Comment: The authors should (briefly) comment on the possible cost-effectiveness (or lack thereof) of such very early (<2 weeks post-AA initiation) biochemical monitoring of patients.

Response: We agree that this is an important area to address but are concerned that including this complex topic will distract from the simple message of the paper. The main aim of early testing is to prevent hyperkalaemia-related death and the cost-effectiveness of this is hard to evaluate. Certainly in UK practice, NICE advocate early testing for patients initiating ACEI/ARB and AA as best practice without consideration of cost.

Reviewer 4 Comments

Comment: The finding that females had a lower odds of being tested in the first 2 weeks is valid, but an odds ratio of 0.9 only realistically highlights a small difference, and while the model presented was "fully adjusted" it would be preferable to also consider what differences in baseline characteristics there were between the genders to regard this finding as more definitive.

Response: We agree and, on reflection, feel that we overemphasised this finding in the original presentation of the paper. Please see the changes we have made in our response to Reviewer 3.

Comment: I wonder if the fact that females had a higher odds of a $>30\%$ increase in creatinine, yet men had a higher odds of a serum creatinine $>220\mu\text{mol/L}$ relates to more than just body muscle mass and perhaps relates to gender differences in the severity of renal function at baseline

Response: Baseline differences in eGFR between genders did exist; a higher proportion of women had baseline eGFRs $<60\text{mls/min}$. However, the models that estimated the odds of $\geq 30\%$ increase in creatinine and odds of creatinine $\geq 220\mu\text{mol/L}$ adjusted for baseline eGFR values. Overall we feel that exploring the potential reasons for these gender differences are beyond the scope of this paper but hope to explore it in our future related work.

Comment: Equally, comparing cessation of treatment rates by gender would help to clarify the insinuation that females may be treated less expeditiously than men without clear justification. Consideration of these aspects would help to backup one of the main conclusions of the paper better.

Response: In response to your earlier point, we agree and have now shifted the emphasis of the paper away from focus on gender-related aspects of monitoring.

Comment: I think it's also worth acknowledging that only 60% of the population had heart failure, and yet the guidelines being presented and used to guide this study's outcomes are focused predominantly on heart failure guidelines. If the conclusion is that there is a need for "better adherence to monitoring guidelines" then either only systolic heart failure patients should be in the population or there should be a discussion point that various guidelines recommend monitoring renal function in different ways.

Response: In the UK primary care coding may not capture all individuals with a diagnosis, so, while 60% of the population were coded to have heart failure, heart failure is likely to be the primary indication for dual ACEI/ARB and AA treatment in a greater proportion. We chose the heart failure guidelines as the structure for the paper as these provide clear advice to GPs (unlike the hypertension guidelines). Our Methods section refers the reader to a supplementary table outlining various different guidelines. However, you are correct that this is an important limitation of the paper and we have therefore added the following to the discussion:

However, to be pragmatic we quantified testing for all patients against the NICE heart failure guidelines, although not all patients will have had cardiac failure as the indication for treatment.

Line 218 Page 15

Comment: In the supplementary table most of the listed guidelines provide no recommendations for frequency of biochemical monitoring.

Response: Thank you for highlighting this. The "x" in the table indicates where the guidelines (columns) require testing according to the frequency laid out in table rows. We have added a footnote to make this clear.

Comment: This leads onto the third point that I think a comment needs to be included regarding the fact that this data is descriptive, and that guideline recommendations are recommendations for clinicians, not mandated.

Response: We agree with this important point and have amended the manuscript in the Methods section as follows:

Guidelines offer a framework for clinicians on how to treat and manage patients but adherence is not obligatory and deviations may reflect appropriate individualised care. Nonetheless, using clinical guidelines to frame this analysis provides a series of time points against which we can quantify frequency of monitoring for patients using combinations of AA and ACEI/ARB.

Line 81 Page 7

Comment: If you are able to demonstrate higher rates of rehospitalisation or higher adverse clinical outcome rates associated with less frequent biochemical monitoring, then the conclusion would be more valid. Similarly a rise in creatinine to $>220\mu\text{mol/L}$ or $>30\%$ may not have any serious implications depending on baseline levels and clinical response. While I completely agree it's extremely useful to know how often people are being biochemically monitored and what their outcome rates are, none of these findings directly demonstrate negligent care, and may possibly (in a more optimistic way) reflect an individualised approach to patient care.

Response: We agree: please see our response to Reviewer 2's second comment.

Comment: Finally its of little surprise that people with higher baseline potassium levels have higher rates of a potassium over 6, or that people with worse eGFRs have higher rates of creatinine >220 - the cutoffs are in guidelines, but are arbitrary.

Response: We agree, but feel that it useful to explicitly understand the odds of an adverse outcomes depending on the baseline values. The fact that higher baseline potassium is not associated with a higher odds of being tested suggests that this message needs to be more broadly understood by practicing clinicians.

Comment: It would be also of research interest to consider the proportional declines in eGFR related to AA therapy, or to follow biochemical outcomes beyond the first post AA blood test to truly consider to clinical impacts of AA treatment for patients. I realise this may extend beyond the primary aims of the paper, but could be considered.

Response: We agree: please see our response to Reviewer 2's second comment.

VERSION 2 – REVIEW

REVIEWER	Jordan Fulcher NHMRC Clinical Trials Centre University of Sydney Australia
REVIEW RETURNED	21-Sep-2017

GENERAL COMMENTS	I'm very happy with the revisions made and think this paper provides important insights into clinical practice. Well done. Two very minor points- 1. In the Adverse biochemical values section, the sentence "A majority of patients with hyperkalaemia discontinued AA within 30 days of the blood test" is a little misleading with a value of 53% - theoretically it's a majority but minimally so, and stating something like "just over half" reads a little differently.2. Could be worth highlighting that after a month only 50% had been checked and just under two thirds at two months, which really sounds concerning no matter what indication the AA was prescribed for, unless there was the possibility blood tests were taken out of area (was this likely?)
--

VERSION 2 – AUTHOR RESPONSE

Comment: I'm very happy with the revisions made and think this paper provides important insights into clinical practice. Well done.

Response: Thank you

Comment: In the Adverse biochemical values section, the sentence "A majority of patients with hyperkalaemia discontinued AA within 30 days of the blood test" is a little misleading with a value of 53% - theoretically it's a majority but minimally so, and stating something like "just over half" reads a little differently.

Response: We have now changed to "approximately half of patients".

Line 186 Page 14

Comment: Could be worth highlighting that after a month only 50% had been checked and just under two thirds at two months, which really sounds concerning no matter what indication the AA was prescribed for, unless there was the possibility blood tests were taken out of area (was this likely?)

Response: We have now highlighted the 2-month statistics in the opening paragraph of the discussion, and in the concluding section.

"Less than two-thirds had blood testing within two months....."

Line 213, Page 16

Line 295, Page 18